# Framework for Indoor Elements Classification via Inductive Learning on Floor Plan Graphs

**Jaeyoung Song and Kiyun Yu \***

Department of Civil and Environmental Engineering, Seoul National University, Seoul 08826, Korea;
sjy9402@snu.ac.kr
\* Correspondence: kiyun@snu.ac.kr; Tel.: +82-10-5068-2628

**Abstract:** This paper presents a new framework to classify floor plan elements and represent them in a vector format. Unlike existing approaches using image-based learning frameworks as the first step to segment the image pixels, we first convert the input floor plan image into vector data and utilize a graph neural network. Our framework consists of three steps. (1) image pre-processing and vectorization of the floor plan image; (2) region adjacency graph conversion; and (3) the graph neural network on converted floor plan graphs. Our approach is able to capture different types of indoor elements including basic elements, such as walls, doors, and symbols, as well as spatial elements, such as rooms and corridors. In addition, the proposed method can also detect element shapes. Experimental results show that our framework can classify indoor elements with an F1 score of 95%, with scale and rotation invariance. Furthermore, we propose a new graph neural network model that takes the distance between nodes into account, which is a valuable feature of spatial network data.

**Keywords:** floor plan analysis; vectorization; graph neural network; indoor spatial data

## 1. Introduction

A floor plan is a drawing that describes the overall layout of a specific level of a building or a structure. There are different ways to format a floor plan, but all floor plans have both structural indoor elements, such as walls, windows, doors, and stairs, and spatial elements, like rooms and corridors, in common. Digitizing floor plans is challenging, since, in most cases, they are basically images without explicit information of any object. Therefore, feature extraction and analysis of indoor spatial data obtained from floor plan images have been done by pre-processing the input image using image-processing techniques and then applying statistical and analytical algorithms. Studies with heuristic algorithms have resulted in high accuracy and precision; however, these algorithms cannot be applied to different types of drawing styles due to the limitations of being dependent on certain types of data [1–6]. To alleviate these limitations, various machine learning-based approaches have been used in floor plan analysis. Among them, Convolutional Neural Network-based approaches have been used the most, as they can be applicable to many styles of floor-plan images. CNN-based approaches only require a basic level of image pre-processing techniques and are robust to floor plan noise. In addition, they can be applied to any style of drawing without the need for transformation, which makes them efficient and versatile [7–10].

However, because these methods perform pixel-level segmentation, the exact shape of indoor elements is hard to capture. To overcome this limitation, these approaches have incorporated additional post-processing steps that abstract the output of the neural network. This, however, results in feature loss of the original indoor elements, such as how polygons are expressed as line vectors. For example, walls should have a thickness and an area of their own; nonetheless, as the shapes get blurry as they pass through the convolution layers, the walls are finally depicted as line vectors by the post-processing algorithms [7,8]. Although abstracting a floor plan layout through machine learning-based

models may be essential for specific user purposes, such as to express navigable areas in IndoorGML format [11], vector outputs that keep the form of the original floor plan image intact can be transformed into various objects depending on the user's purpose, owing to the high flexibility and deformation ability of the vector data type.

In this paper, we propose a framework that finds any kind of element in the floor plan without losing the shape information. It first vectorizes the input floor plan image as it is to maintain the shape of the original indoor elements and minimize the abstraction. The polygon vector set is then converted into a region adjacency graph. The graph is then fed to an inductive learning-based graph neural network (GNN), which is used to compare multiple floor plan graphs and perform node classification by analyzing inherent features and the relationships between the nodes. This allows the user to classify basic indoor elements (e.g., walls, windows, doors, etc.) and symbols, together with space elements (e.g., rooms, corridors, or outer spaces), without losing their shape and arial features. Furthermore, a new GNN model, the Distance-Weighted Graph Neural Network (DWGNN), is presented. In accordance with the first law of geography [12], the neighboring nodes that are close to a target node should be given relatively high attention values compared to the neighbors that are far apart from the target node. To do so, we developed a GNN model that assigns attention values to the neighbors in a target node's neighboring subgraph. The DWGNN considers the distance information between the nodes, expressed by edge features, in the spatial network (graph). To evaluate the performance and expressiveness of the new proposed floor plan analysis framework, we applied it to two floor plan datasets and one data-augmented dataset.

The remainder of the paper is structured as follows. In Section 2, we discuss the limitations of previous researches related to floor plan analysis, particularly regarding indoor element classification using rule-based methods and machine learning approaches. In Section 3, based on the described limitations, we propose a framework for floor plan element classification via GNN. Finally, we analyze the results on three datasets and discuss issues and further research.

## 2. Related Works

### 2.1. Rule-Based Heuristic Methods and Machine Learning Algorithms in Floor Plan Analysis Research

Detecting and classifying floor-plan basic elements or regions have been studied for many years, with various approaches. Ruled-based heuristic approaches utilize methods based on image processing, such as the morphological filtering [1,6], Hough transformation [2,4], text/graphic recognition [3,4], or using graph algorithms [5,13]. Although they have showed meaningful outputs, rule-based heuristic approaches struggle to maintain the shapes of elements, and can only be applied to specific drawing styles.

To avoid these style-dependent heuristics and take expressive generality among various drawing styles, approaches using machine learning algorithms have emerged . De las Heras et al. [7] utilized a machine learning algorithm to detect indoor elements, and then converted the output into vector data. Citing the limitation that the existing rule-based methods are ad hoc and only applicable to certain drawing styles, they presented an automatic method that detected room boundaries in floor plans invariant to the style of the drawings. They used a Support Vector Machine Bag of Visual Words (SVM-BOVW) to detect the pixel boundaries of the structural elements, which included walls, doors, and windows, and then created the vector data. In addition, the model recognizes room boundaries in the floor plan by finding closed regions surrounded by vectors of structural elements. Liu et al. [8] trained a CNN to detect the junctions, such as wall corners, in a floor plan and applied integer programming to extract vector data by combining the junctions to build simple primitives like walls and windows. In addition, they found spaces with closed combinations of simple primitives. However, all of the elements were assumed vertical and horizontal, thus failing to secure the shapes of the elements and resulting in largely abstracted primitives, such as expressing the walls with line vectors. Dodge et al. [9] used Fully Connected Networks (FCN) and Faster R-CNN to segment walls and detect

objects, respectively, in floor plans with various drawing styles. They also used OCR to be able to recognize the size of the rooms and to place furniture models scaled to the scene. Zeng et al. [10] proposed a method that detects and classifies walls, doors, windows, and rooms by training a VGG encoder-decoder. Unlike [8], their method is applicable to non-rectangular shape elements and is able to obtain the shape features of indoor elements. In addition, they used an attention mechanism for the decoder units. The two decoders share the attention values to predict the boundary and the type of rooms. However, their method is limited to only a few classes, which are used as a layout to help the decoder find the room boundaries; these boundaries are ultimately placed under the same class.

Floor plan analysis using machine learning algorithms has shown great potential on various floor plan datasets, but still, each approach has its own limitations and shortcomings. Models trained on various input floor plan datasets may have great adaptability, but their outputs may be blurry as they perform pixel-level segmentation. This creates problems in the output, such as unconnected lines, which result in unclosed vectors. In many cases, room detection and recognition depend heavily on structural elements in the floor plan, such as walls, doors, and/or windows, and if these structural elements have unclosed issues, they will considerably affect the room formation process. Because elements may lose their shape information during the vectorization process [7,8], some approaches omit this process in order to secure the shape features [9,10]. In addition, none of the approaches that concentrated on detecting structural elements and space elements considers symbolic elements, such as cabinets, baths, or toilets, among others.

### 2.2. Graph Neural Network (Gnn) and Floor Plan Analysis Using Gnn

A graph data structure consists of a finite set of nodes (vertices) and edges (links). A node represents an entity, and an edge represents a relation between two nodes. Graphs are often referred to as non-euclidean data structures, since they are not confined to any particular dimension. Existing deep learning algorithms applied to euclidean data structures have shown great performance. However, existing deep learning models are unable to learn graphs because permutation between nodes can appear in various ways. Accordingly, GNNs [14,15] have been devised to describe a way to express the order of the nodes and allow the neural network to learn graph data structure.

In recent years, GNNs have undergone numerous variations of the basic definition. Kipf et al. [16] introduced the graph convolution networks (GCN) to utilize the convolution operation on graphs by updating the nodes' latent vector using a normalized Laplacian matrix as an adjacency matrix of the input graph. Hamilton et al. [17] proposed Graph-SAGE and showed that the results of the latent vector of outcome differs with various AGGREGATE functions, and applied this notion to perform inductive learning to train the model with, not a single, but multiple graphs. Xu et al. [18] found that GNN models cannot be properly trained, and introduced a new model, the graph isomorphism network (GIN), that can perform as much as the WL test, which is an isomorphism test for graph structure. They also classified graph-related tasks that can be appropriately applied according to the AGGREGATE methods.

A GNN can analyze various real-world problems. Due to their inherent characteristics, they can be represented as a graph, and GNN would take them as input to analyze and predict. A floor plan can also be converted into a graph by treating cell regions as nodes and constructing an adjacency matrix based on the adjacency among the regions of the floor plan. Various graph algorithms and analyses have been applied to floor plan graphs. In particular, floor plan graphs have been extensively used in the field of floor plan design research, which has recently studied different methodologies using GNN. For example, an automated generation framework for floor plan design using GNNs was proposed by Hu et al. [19]. When it comes to detecting and classifying the indoor symbols or elements, Renton et al. [20] applied GNN to classify symbols in the floor plan. They pre-processed floor plan images and considered the centroids of regions surrounded by black pixels as nodes. A region adjacency graph is then constructed by connecting the nodes that share a

pixel line. Then, the floor plan graph is fed into a GNN model as the input graph and a graph is obtained where the nodes were classified according to their local dependencies. This study is the first to use GNN to classify symbols in floor plan images. However, it only targeted symbols and objects, excluding walls and rooms, which are the most important elements of floor plan images. In addition, the final output of the approach is limited to graphs that represent only the symbol classes and are not converted into vector-format output for utilization.

## 3. Materials and Methods

To overcome the limitations described in the previous studies on floor plan element extraction and classification tasks, the following requirements were defined.

1. The framework must detect and classify space elements, such as rooms, together with basic elements (walls, doors, etc.) and symbols.
2. The framework must start with raster data and output vector data maintaining shape without abstraction.
3. The framework must perform inductive learning by separating a set of graphs of various types and sizes into graph units, rather than transductive learning that deals with a single large graph.

To meet these requirements, developing and extending the ideas used in [20], we propose a new framework as follows. The raster floor plan image is fed to the framework as the input data. The image is first pre-processed in order to obtain a binarized image to be vectorized. The closed regions in the image become polygons after the vectorization process. The polygons with shape features are then converted into a region adjacency graph (RAG) according to their adjacent relationship with neighboring polygons. The RAG is then fed into the neural network to train the GNN model. The final output of the framework is a set of polygons with different classes. The overview of the proposed framework is shown in Figure 1.

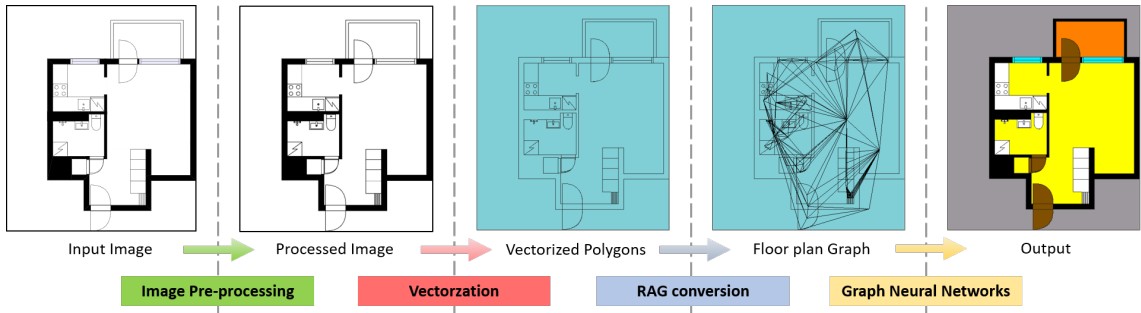

**Figure 1.** Overview of the proposed framework. The input floor plan image is pre-processed to erase texts and get binarized. The processed image is then vectorized depending on its closed regions and converted to an RAG. The floor plan graph is input to a GNN module in order to classify each polygon according to its and the neighbors' feature vectors.

### 3.1. Image Pre-Processing and Vectorization

The pre-processing phase may vary depending on the layout style of the floor plans, but most consist of text removal and binarization. The three channels of the input floor plan image (Red, Blue, and Green) are merged into a single channel and binarized. The text information is removed using the OCR algorithm. The processed image is then vectorized. De [6] assumed that only walls are depicted as thick black lines in a floor plan layout; therefore, thick and thin lines can be distinguished using a morphological transformation, and thick lines can be considered as walls. However, this approach can only be applied to specific floor plan styles, as in many cases, walls could be represented as white areas. To vectorize the image regardless of the floor plan drawing style, we chose to vectorize the white and black areas separately.

The detailed process is described as follows. A closed area surrounded by black pixels in the image becomes a polygon object. Likewise, a set of polygons is generated from all the closed white areas in the plan (Figure 2c). If the floor plan layout contains black areas, the empty polygon with the size of the floor plan (Figure 2b) does the difference operation on the white polygon set. This creates a second set of polygons that represent the black areas in the floor plan (Figure 2d). Since we binarized the image, there are only two colors in the image, which make it possible to turn every area in the image into a polygon regardless of the drawing or layout style. Lastly, the two polygon sets get merged, and the complete set of polygons is generated (Figure 2e). During this process, the regions occupied by the pixel lines that surround the polygons will not be included in the polygons. Therefore, the polygons will be buffered by the thickness of the pixel line before executing the difference operation (Figure 2f). Buffering the polygon is crucial because, if the polygons are separated from one another, the adjacency operation will return false when constructing the adjacency graph. Taking the thickness of the pixel line $t$, the buffering distance parameter is selected as $t/2$, as each pixel line has to be covered by two polygons from two directions.

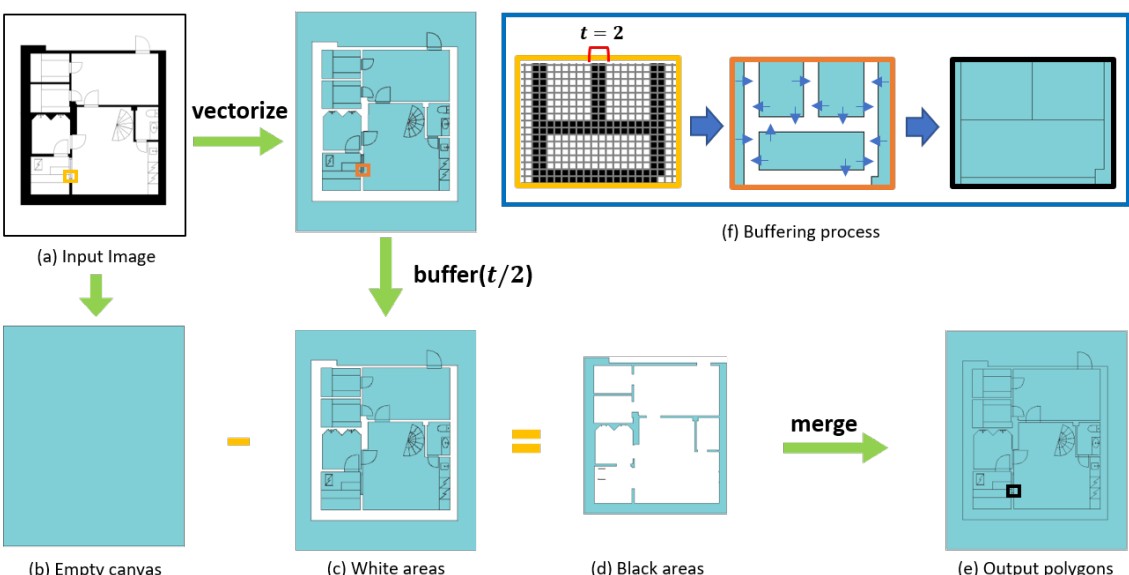

**Figure 2.** Overview of the vectorization process. The white areas in (**a**) are vectorized and buffered according to the thickness of the pixel lines surrounding them (**c**). The black areas are converted into polygons (**d**), which are generated by the difference operation between (**b**,**c**). Finally, the complete polygon set (**e**) is generated by merging the two polygon sets. (**f**) describes the detailed process of polygon buffering (the frame color of each step shows the respective small square in the process in detail).

### 3.2. Region Adjacency Graph (Rag) Conversion and Feature Extraction

Algorithm 1 describes the RAG conversion process. First, an empty graph $G$ is created, and for each polygon element $p$ in the polygon set $P$, the polygon centroid of $p$ ($v_p$) is added as a node. To construct the edge set of $G$, $p$ executes an INTERSECTS operation on another polygon $q \in P, q \neq p$. With the rest of the polygon elements in $P$, $p$ would need to execute the INTERSECTS operation $|P| - 1$ times, and the number of iterations for $P$ would increase exponentially with the number of nodes. To reduce the number of iterations and the complexity, instead of two nested loops, we used an STRtree [21], which is a spatial indexing algorithm based on an R-tree. The tree returns a resulting polygon set $Q$ when $p$ queries the INTERSECTS of the other spatial objects. If a polygon element $q$ is in $Q$ and $q$'s area is bigger than the minimum area parameter $m$, the edge between $v_p$ and $v_q$ is added to the edge set $E$. By using the STRtree, the time complexity of the RAG conversion process is reduced from $O(n^2)$ to $O(n \log_m n)$. $n$ is the number of polygons (nodes) and $m$ is the number of entries in the tree.

---

**Algorithm 1:** RAG conversion

---

**input** : A polygon set $P$, a minimum area parameter $m$
**output:** A floor plan graph $G$

```
// Create a graph with adding polygon nodes
```
1   $G \leftarrow$ Graph()
2   **for** $p \in P$ **do**
3      $v_p \leftarrow p$.centroid
4      $G$.addNode($v_p$)
```
// Build STRtree with the polygon set
```
5   $tree \leftarrow$ STRtree($P$)
6   **for** $p \in P$ **do**
```
// Query intersects function with polygon p
```
7      $Q \leftarrow tree$.INTERSECTS($p$)
8      **for** $q \in Q$ **do**
9          **if** $q \neq p$ **and** $q$.area $> m$ **then**
10              $G$.addEdge($v_p, v_q$)

11 **return** $G$

---

The constructed graph $G = (\mathcal{V}, \mathcal{E})$ consists of the node set $\mathcal{V}$ and the edge set $\mathcal{E}$, which represent the adjacent relationship among nodes in the floor plan layout. A polygon node $v_p$ is recognized as the centroid of $p$ and has its own unique feature vector $\mathbf{x}_{v_p} \in \mathbf{X}_v$. $\mathbf{X}_v$ is the feature matrix of $G$ whose size is the number of $\mathcal{V}$ and the dimension of the node feature vector $d_v$. $e_{pq}$ is an element of the edge set $E$, which represents how the polygon nodes $v_p$ and $v_q$ are connected to each other. An edge also has its own feature vector $\mathbf{x}_{e_{pq}} \in \mathbf{X}_e$. Each edge feature vector has $d_e$ features. If $d_e = 1$, we consider the edge feature as the weight value between two nodes. The constructed RAG $G$ is described as follows:

$$G = (\mathcal{V}, \mathcal{E}, \mathbf{X}_v, \mathbf{X}_e). \tag{1}$$

In the framework, we used four features for $\mathbf{X}_v$ and a single feature for $\mathbf{X}_e$ (a weight value). A node feature vector for node $v_p$ ($\mathbf{x}_{v_p} \in \mathbf{X}_v$) consists of the area of $p$, the degree of the node, the normalized central moment of order 1 and 1 for the polygon, and the Zernike moment [22] of order 4 and repetition 2 ($\mathbf{x}_{v_p} \in \mathbb{R}^4$). The two used moments are scale- and rotation-invariant. The edge feature vector $\mathbf{x}_{e_{pq}} \in \mathbf{X}_e$ consists of the euclidean distance between its two nodes ($v_p, v_q$). Edge features are considered as weights of $G$ since the edge feature dimension parameter $d_e = 1$. The polygon set $P$ and the RAG $G$ are constructed for each floor plan layout in the datasets. In the following section, we will describe various GNN models to classify the classes of polygons in $P$ using $G$.

### 3.3. Graph Neural Network Models

A GNN performs a prediction on various tasks, such as node classification, edge prediction, and graph classification. Like other deep learning models, it extracts a unique embedding vector of each entity in the target dataset and compares its similarity to other embedding vectors to predict a result as close as possible to the label data. The domain of interest of GNN varies, including nodes, edges, graphs, and subgraphs [23]. The GNN takes the adjacency matrix $A$ and the feature matrix $X$ of the target graph as input. $A$ represents the relationship between the nodes, and $X$ holds the feature vector for each node in the target graph. If features are found on the edges, they can be added to the value for $A$ or taken as a separate edge feature matrix.

GNN has multiple layers, and each layer consists of the AGGREGATE and UPDATE functions. The AGGREGATE function aggregates information coming from the neighboring nodes and returns a message. The UPDATE function combines the target node's

embedding vector and the message to update the new latent embedding vector of the target node. This process is called message passing. The forward-propagation process of a vanilla GNN model for generating the new embedding vector of node $v$ at layer $k$ can be as follows [24]:

$$\mathbf{h}^k_{\mathcal{N}(v)} = \text{AGGREGATE}^k\left(\{\mathbf{h}^{k-1}_u, \forall u \in \mathcal{N}(v)\}\right)$$
$$\mathbf{h}^k_v = \text{UPDATE}^k\left(\mathbf{h}^{k-1}_v, \mathbf{h}^k_{\mathcal{N}(v)}\right),$$

(2)

where $\mathcal{N}(v)$ is the set of neighboring nodes of $v$ and $\mathbf{h}^{k-1}_u$ is the latent embedding vector of $u \in \mathcal{N}(v)$ at layer $k-1$. AGGREGATE$^k$ aggregates the embedding vectors to return the message $\mathbf{m}^k_{\mathcal{N}(v)}$. UPDATE$^k$ takes $\mathbf{m}^k_{\mathcal{N}(v)}$ with $\mathbf{h}^{k-1}_v$, which is the embedding vector of node $v$ at layer $k-1$ as input and generates the embedding vector of node $v$ at layer $k$. Both AGGREGATE$^k$ and UPDATE$^k$ are arbitrary differentiable functions at layer $k$ (i.e., neural networks). These two functions can be defined in various ways depending on the task the model wants to solve. The definition of the AGGREGATE function allows neighboring nodes to determine how they will affect the target node, and the UPDATE function determines how to combine the message and target node's embedding vector of the previous layer, and how the embedding vector is generated.

Our goal is to classify the polygon nodes by extracting the latent embedding vectors for each node in the floor plan graph, which is categorized as a node classification task. The performance of a GNN model for node classification highly depends on the structure of its network, not only regarding the functions used for AGGREGATE and UPDATE, but also regarding the number of layers. As the number of layers increases, the wider the neighborhood node information is included. This is similar to the receptive field of a target pixel in a CNN; as the number of layers increases, the receptive field widens.

### 3.3.1. A GNN Variant for Inductive Learning on Graphs

Most of the GNN models target one large graph, such as a social network, focused on generating embedding nodes from a single fixed graph. However, from a real-world application point of view, a GNN model that generates embedding vectors for unseen nodes, or entirely new graphs, is needed [17]. Figure 3 explains the difference between transductive learning and inductive learning in graphs. Our study also required the inductive learning GNN model as the floor plan datasets mostly consist of various floor plans, and each floor plan is converted into a unique graph. Inductive learning enables prediction on these completely unseen graphs. We trained the inductive learning-based GNN model on the floor plan graphs of the training set, and the model predicted the classes of the nodes in the test-set floor-plan graphs.

Many existing spatial-based GNN models are transductive learning-based GNN models [16,18], while GraphSAGE [17] is based on inductive learning. GraphSAGE is a general inductive framework for generating latent embedding vectors of completely unseen nodes. In the GraphSAGE model, which consists of $K$ layers, the algorithm for generating an embedding vector of node $v$ at layer $k$ is as follows:

$$\mathbf{h}^k_{\mathcal{N}(v)} = \text{AGGREGATE}^k\left(\{\mathbf{h}^{k-1}_u, \forall u \in \mathcal{N}(v)\}\right)$$
$$\mathbf{h}^k_v = \sigma\left(\mathbf{W}^k \cdot \text{CONCAT}(\mathbf{h}^{k-1}_v, \mathbf{h}^k_{\mathcal{N}(v)})\right),$$

(3)

where $\mathbf{W}^k$ is a weight parameter matrix to be trained and $\sigma$ is a non-linear activation function (e.g., sigmoid function). The UPDATE function in GraphSAGE is a concatenation function multiplied with the weight matrix.

The initial vector for node $v$ is the input node feature vector, and, as the number of layers increase, the embedding vector of node $v$ holds the information coming from farther neighbors. This means that, if $k = 0$, $\mathbf{h}^0_v$ is $\mathbf{x}_v \in \mathbf{X_v}$, and $\mathbf{h}^K_v$ aggregates all the information of the neighbors within $K$-hops from $v$ in the graph. Hamilton et al. [17] showed the

difference of performance among various AGGREGATE functions. For the AGGREGATE function, they used the MEAN operator (similar to GCN [16]), an LSTM layer, and a POOL function based on the MAX operator with a weight matrix parameter. Unlike others, LSTM is not permutation-invariant, but shows strong performance and expressiveness as it trains additional neural networks [17].

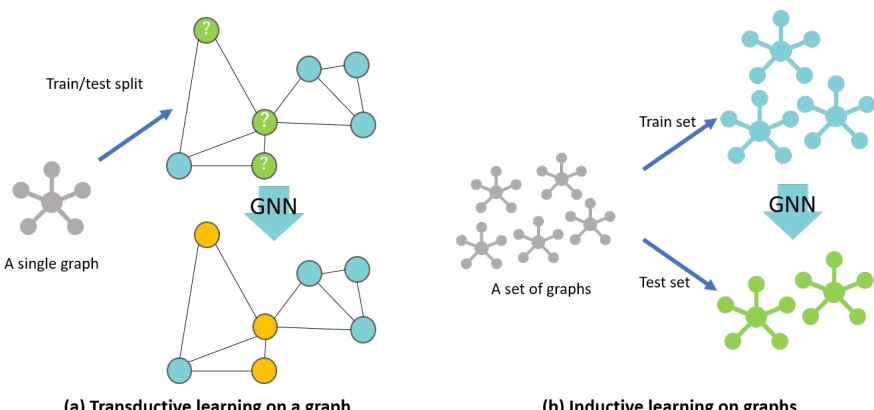

(a) Transductive learning on a graph      (b) Inductive learning on graphs

**Figure 3.** Node classification on a transductive learning GNN method (**a**) and on an inductive learning GNN method (**b**). In the transductive learning method, (**a**) the model is trained by accessing all the nodes and edges in order to predict the class of nodes in the test set (denoted by question marks). In the inductive learning method (**b**), on the other hand, the set of graphs is split into training and test set, and the test set is predicted with a GNN model trained on a set of training graphs.

### 3.3.2. A GNN Model to Utilize Distance Weight Feature

A graph describing a real-world example may not only have node features, but also edge features. In spatial networks, the distance between two nodes can be expressed as an edge feature or the weight value of the graph [25]. The edge weight values are an important feature in that they describe the relationship between nodes in a spatial graph. Under the first law of geography, neighboring nodes that are close to a target node should be given relatively high attention values compared to the other neighbors that are far apart from the target node [12].

However, most of the existing GNN models do not leverage the edge feature in their networks. Studies that have utilized the edge feature in node and graph classification tasks have focused on multi-dimensional features, not single-dimension features like weight values in spatial networks [26,27]. Glimmer et al. [26] proposed a model utilizing edge features in the message-passing process. However, their model is too general, since the message function $M_t$ is not a specific method and could be any function. A GNN model that can handle spatial networks consisting of nodes and distance weights is thereby needed.

We propose a new inductive learning-based GNN model named the Distance-Weighted Graph Neural Network (DWGNN). DWGNN is a GraphSAGE-based model in which an edge feature mechanism is applied in the message-passing process. Its target graph represents a spatial network where the distance between nodes is a one-dimensional weight value. When DWGNN aggregates the neighbor's information, it assigns the attention values to neighboring nodes' embedding vectors according to the relative distance from the target node. The update process of DWGNN is as follows.

$$\mathbf{h}^k_{\mathcal{N}(v)} = \text{AGGREGATE}_k\left(\mathbf{W}^k_0 \cdot (\mathbf{h}^{k-1}_u \odot \text{softmin}(\mathbf{e}_{\mathcal{N}(v)})_u), \forall u \in \mathcal{N}(v)\right)$$
$$\mathbf{h}^k_v = \sigma\left(\mathbf{W}^k_1 \cdot (\mathbf{h}^{k-1}_v + \mathbf{h}^k_{\mathcal{N}(v)})\right),$$

(4)

where $\mathbf{e}_{\mathcal{N}(v)}$ is the distance weight vector of node $v$ and its neighboring node set $\mathcal{N}(v)$ and $\odot$ denotes element-wise multiplication. Softmin is a function that converts every element of $\mathbf{e}_{\mathcal{N}(v)}$ into an attention value. It is defined as follows,

$$\text{softmin}(x_i) = \frac{exp(-x_i)}{\sum_j exp(-x_j)}. \tag{5}$$

Similar to the softmax function, which converts each element of the input vector to a value between $[0, 1]$ and the sum of all converted values is equal to 1 such as a probability value, the softmin function returns a normalized vector where each element gets a larger attention value if its weight value is relatively smaller than others. This assigns nearby neighboring nodes a larger attention value compared to those far apart. In addition, similar to GraphSAGE, the AGGREGATE function of DWGNN can be chosen between various functions, such as SUM, MEAN, MAX, and LSTM. The update process of DWGNN is shown in Figure 4. If the weights play a significant role in a spatial network, DWGNN can be an appropriate GNN model to analyze such graphs.

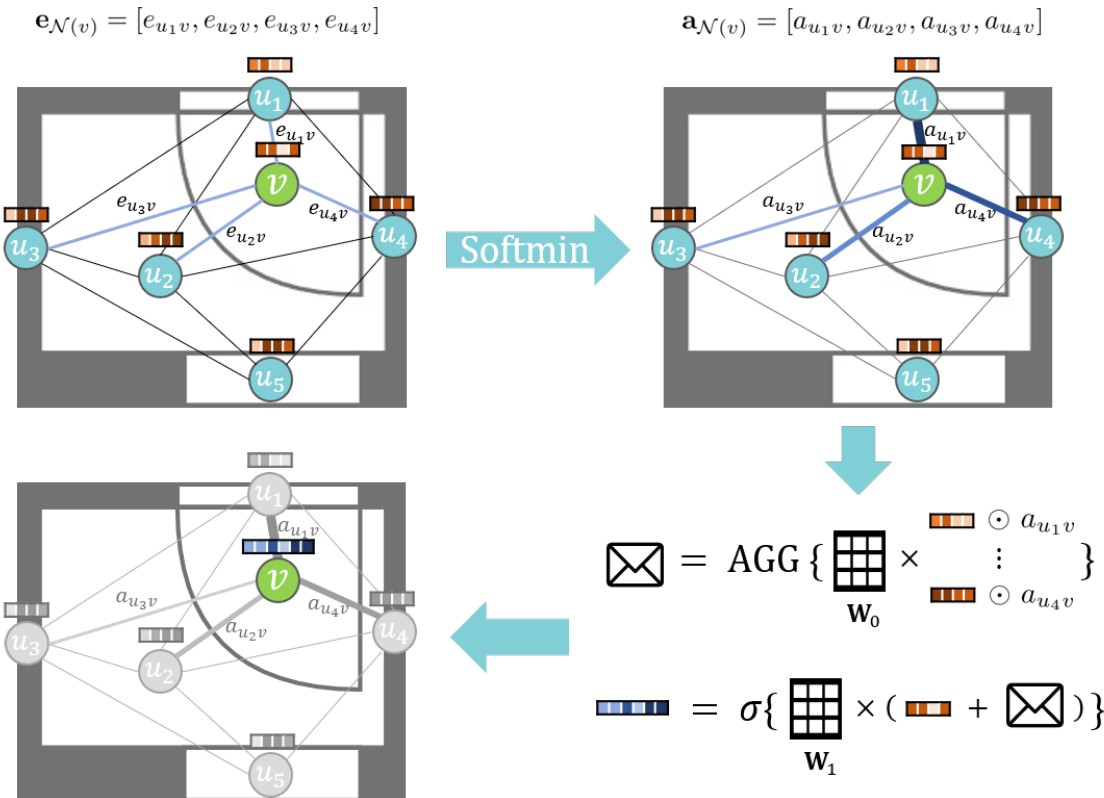

**Figure 4.** Visual illustration of the update process of node $v$ (a door segment). The softmin function assigns respective attention values to each neighbor of $v$ according to their distance to $v$ ($e_{u_i v}$). Each node's embedding vector at layer $k-1$ is element-wise multiplied with a respective attention value. They pass through a weight matrix $W_0$ and aggregated to a message. This message is added to $v$'s embedding vector at layer $k-1$ and multiplied with the weight matrix $W_1$. The result is the embedding vector of node $v$ at layer $k$. In the Figure, $\mathbf{a}_{\mathcal{N}(v)}$ is the converted attention vector and **AGG** is the AGGREGATE function.

## 4. Results

### 4.1. Datasets

To test and evaluate the proposed framework, we conducted experiments on two different floor plan benchmarks, together with one data-augmented dataset. We did not use the floor plan datasets which had been used in previous works, since their raster images had a lot of noise and/or the resolution was too low (e.g., R2V [8], RF-P [9]) or unable to be

obtained (ILPIso [20]). We will discuss applicability issues in detail in Section 5. In what follows, we will address two different floor plan datasets we used in the experiments. Both datasets consist of basic structural classes and spatial element classes together with the object class. Object class is comprised of various furniture and installations placed in an indoor environment, such as cabinets, chairs, or toilets. Any other object not in a structural or spatial category will be assigned to the object class.

The CubiCasa5K [28] (CubiCasa) dataset consists of 5000 different apartment floor plans. The quality of the floor plan images varies from clean, noise-free ones to scribbled or noisy ones. They are divided into three categories: high quality, architectural high quality, and colorful. We used the SVG formatted labeled floor plan images hand-annotated by experts as input data, by converting them into raster image data. After we vectorized the polygons, we classified the polygons into eight classes: four structural element classes (walls, windows, doors, and stairs), three spatial element classes (rooms, porches, and outer-space), and the object class comprised of various symbols. We selected the 400 high-quality floor plan images and split them evenly into training and test sets.

The University of Seoul (UOS) dataset containing plans for seven floors of the 21st century building at the University of Seoul, was used to evaluate whether the framework is applicable to large-area floor plan data along with relatively small ones, such as Cubi-Casa5K. We exported the CAD floor plan data into raster data. We classified the elements of vector plans into nine classes: five structural element classes (including elevators), three spatial element classes (rooms, corridors, and X-rooms), and the object class. Though the number of plans is limited because of security issues, if the framework is able to generalize and classify the indoor elements in UOS, we can say that the framework works well with a smaller number of floor plans. We used a seven-fold cross-validation strategy. Each session consisted of six training plans and one plan for the test. The final result was averaged by all seven sessions.

### 4.2. GNN Models

We implemented four GNN models for performance comparison. We conducted inductive learning experiments under the same conditions and settings. The following are the used GNN models.

1.  GCN [16]: Graph Convolution Networks aggregate the neighbor nodes of the target node using a symmetric normalized graph Laplacian $\tilde{\mathbf{D}}^{-\frac{1}{2}}\tilde{\mathbf{A}}\tilde{\mathbf{D}}^{-\frac{1}{2}}$ made with a self-loop adjacency graph $\tilde{\mathbf{A}} = \mathbf{A} + \mathbf{I}$ and a diagonal degree matrix $\tilde{\mathbf{D}} = \sum_j \tilde{\mathbf{A}}_{ij}$. The embedding vectors of the target nodes are generated by summing the information of neighboring nodes and projecting onto a weight matrix. The update process of GCN is

$$\mathbf{h}_v^k = \sigma\big(\mathbf{W}^{k-1} \cdot \sum_{u \in \mathcal{N}(v)} \frac{1}{c_{vu}} \mathbf{h}_u^{k-1}\big), \tag{6}$$

where $c_{vu}$ is a normalization constant for the edge $(v, u)$ originating from $\tilde{\mathbf{D}}^{-\frac{1}{2}}\tilde{\mathbf{A}}\tilde{\mathbf{D}}^{-\frac{1}{2}}$.

2.  GIN [18]: A Graph Isomorphism Network was proposed to maximize the discriminative and representational power of each node in a graph. It shows almost the same performance as the Weisfeiler–Lehman graph isomorphism test [29]. We used MAX, MEAN, and SUM operations as the AGGREGATE function in our experiments. The update process of GIN is

$$\mathbf{h}_v^k = \sigma\bigg(\text{MLP}^k\Big((1 + \epsilon^k) \cdot \mathbf{h}_v^{k-1} + \text{AGGREGATE}\big(\mathbf{h}_u^{k-1}, u \in \mathcal{N}(v)\big)\Big)\bigg), \tag{7}$$

where $\text{MLP}^k$ is a multi-layer perceptron placed at layer $k$ to maximize the discriminative power of the generated embedding vectors. Along with MLPs, $\epsilon^k$ is a scalar parameter at layer $k$ to be trained. We fixed $\epsilon^k = 0$.

3. GraphSAGE [17]: We used the same model as introduced in Section 3.3.1 MEAN was excluded from the experiment because it does not differ much from the propagation rule of GCN. When using the POOL aggregator, a weight matrix was added prior to the MAX operation to increase the expressive power of the message function. The POOL aggregator is defined as follows:

$$\text{AGGREGATE}_k^{\text{pool}} = \max\big(\{\sigma(\mathbf{W}_{\text{pool}}^k \mathbf{h}_u^k + \mathbf{b}), \forall u \in \mathcal{N}(v)\}\big). \tag{8}$$

4. DWGNN: The model developed by the authors and introduced in Section 3.3.2 was implemented. MAX, MEAN, SUM, and LSTM were used for the AGGREGATE function in our experiment.

### 4.3. Implementation Details

In our experiment, each floor plan image of the datasets was vectorized and labeled according to the class conditions described earlier. The parameters used in the vectorization process were the minimum area parameter $m$ as 20 and $t$ as 2. All the node and edge features in the graphs were scaled using the standardization technique. To train the GNN models, we used the Adam optimizer with an initial learning rate of 0.01. Batch normalization [30] was applied to every hidden layer for CubiCasa. The number of hidden layers was six for every GNN model, and the MLPs had two layers for the GIN [31]. The hyper-parameters for experiments were: (1) The number of hidden dimensions for the hidden layers was fixed to 128; (2) for CubiCasa, mini-batches of 10 graphs were set for each iteration and no mini-batches were set for UOS, as we used cross-validation strategy for it; (3) the number of epochs was set to 1000 for all GNN models except inductive learning-based models with a LSTM aggregator (set to 300). Since the LSTM has more parameters to train, the epochs of the inductive learning-based models with a LSTM aggregator was set lower than that of other models.

The hardware characteristics used for the experiments were an Intel i7-9700KF CPU, an NVIDIA GeForce GTX 1660 Ti GPU, and 64 Gb of RAM. For the code implementation, we used the Rasterio package for vectorization and the Shapely, GeoPandas, NetworkX packages for the creation and management of polygon vectors and graphs. GNN models were built using the Deep Graph Library [32] with PyTorch backend. The code is available at https://github.com/LymanSong/FP_GNN (accessed on 22 February 2021).

### 4.4. Experiment on the Cubicasa Dataset

Table 1 shows the results of the predicted classes of elements in the CubiCasa test set using different GNN models and aggregate methods. Among the GNN models, Graph-SAGE showed the highest accuracy. In addition, the LSTM aggregate method showed the highest results.

The accuracy for stairs was relatively low in all models. This is because, given that stairs are depicted as a set of rectangular polygons, rectangles often appear in different elements' classes. In addition, stair polygons with different shapes share one single class, and the number of plans including stairs is significantly lower. On the other hand, windows and doors have high accuracy rates, apparently because each of them share a highly defined structure shape in the drawing style of CubiCasa.

We can find that, compared to the transductive learning-based models (GCN and GIN), the inductive learning-based models (GraphSAGE and DWGNN) performed well on recognizing spatial elements. In Table 1, DWGNN with the SUM method slightly underperformed compared to GIN with the SUM method, but in the case of spatial elements (rooms, porches, and outer spaces) it classified better than GIN with SUM. If we divide the element classes into two classes (spatial and non-spatial) the inductive learning-based models found the spatial classes much better than the transductive learning-based models did. This means that inductive models can generalize the characteristics of classes well and

easily find the dominant features on unseen data, such as predicting whether it is spatial or non-spatial by looking at the area attribute.

**Table 1.** Class-wise accuracy comparison by different GNN models on the CubiCasa dataset (micro-averaged F1 score). AGG stands for the AGGREGATE method.

| GNN Model | AGG | Objects | Wall | Window | Door | Stair | Room | Porch | Outer Space | Overall |
|---|---|---|---|---|---|---|---|---|---|---|
| GIN | MEAN | 0.9001 | 0.8009 | 0.9176 | 0.8029 | 0.5453 | 0.8092 | 0.6719 | 0.7879 | 0.8577 |
| GCN | · | 0.9113 | 0.8118 | 0.9142 | 0.8154 | 0.5398 | 0.8325 | 0.54 | 0.7453 | 0.8658 |
| GIN | MAX | 0.9241 | 0.8842 | 0.9454 | 0.8816 | 0.5968 | 0.8833 | 0.75 | 0.7849 | 0.9025 |
| DWGNN | MEAN | 0.9392 | 0.8485 | 0.9367 | 0.8942 | 0.7653 | 0.903 | 0.6982 | 0.9038 | 0.9137 |
| DWGNN | MAX | 0.9429 | 0.8571 | 0.9456 | 0.898 | 0.7854 | 0.9133 | 0.7215 | 0.9048 | 0.9201 |
| DWGNN | SUM | 0.9441 | 0.8648 | 0.9428 | 0.9054 | 0.7612 | 0.9164 | 0.7268 | 0.9119 | 0.9214 |
| GIN | SUM | 0.9445 | 0.8991 | 0.9783 | 0.9063 | 0.6572 | 0.9067 | 0.7352 | 0.8664 | 0.9283 |
| DWGNN | LSTM | 0.9597 | 0.9224 | 0.971 | 0.94 | 0.7913 | 0.9313 | 0.7849 | 0.9233 | 0.9471 |
| GraphSAGE | POOL | 0.9586 | 0.9157 | 0.9765 | 0.941 | 0.7675 | 0.9377 | 0.8449 | 0.9289 | 0.9478 |
| GraphSAGE | LSTM | **0.9708** | **0.9466** | **0.9896** | **0.9557** | **0.8341** | **0.9617** | **0.8832** | **0.9625** | **0.9651** |

Figure 5 shows the results of visualizing examples of floor plans analyzed through the proposed framework. The framework first vectorizes the input images and converts them into RAGs. The trained GNN models then take these graphs as inputs and extract features to predict the classes of polygons. Compared to the ground truths, inductive learning-based models can classify the basic classes and spatial elements well. On the other hand, the transductive learning-based models fail to predict some basic and spatial element classes. In particular, GCN and GIN were unable to find the doors and walls correctly. As stated earlier, all models classified the stairs incorrectly.

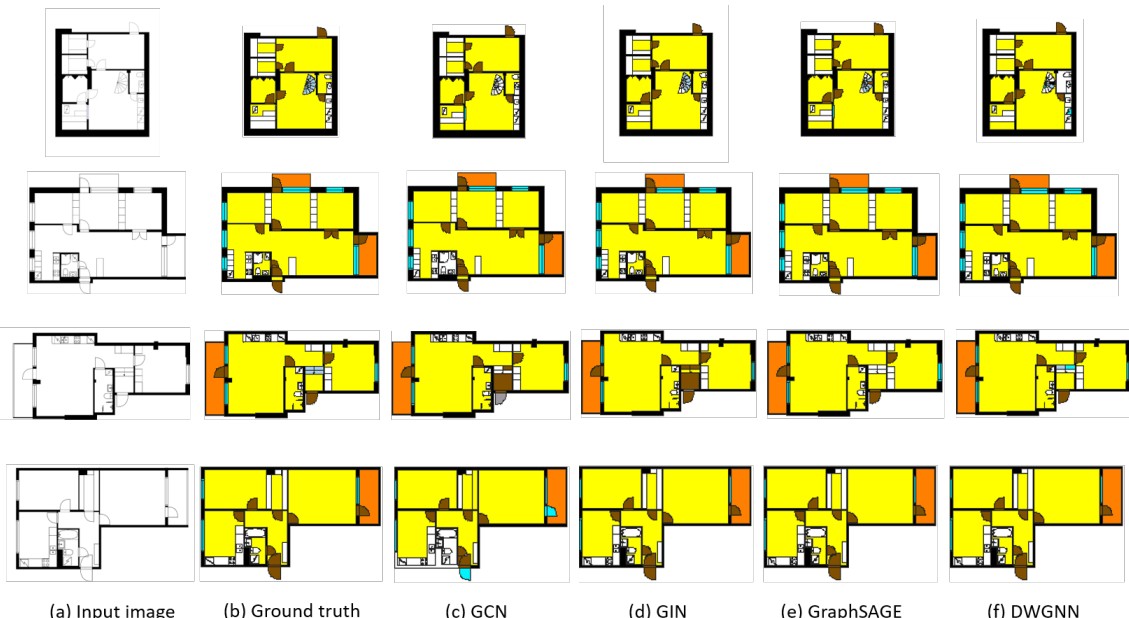

| (a) Input image | (b) Ground truth | (c) GCN | (d) GIN | (e) GraphSAGE | (f) DWGNN |

**Figure 5.** Examples of input image (**a**) and ground truth (**b**), and visual comparison of indoor element classification results by GNN models for transductive learning (**c**,**d**) and inductive learning models (**e**,**f**). The element class "outer space" is erased for visibility.

## 4.5. Experiment on Large and Complicated Floor Plans: Uos and Uos-Aug

Small-area floor plans have fewer polygons and their RAGs have a relatively simple structure compared to large and complex buildings. We conducted experiments on large and complicated floor plans to test our framework. The floor plans of the UOS dataset

were large and complicated, thus resulting in many polygons with complex relationships. The number of floor plans in the UOS dataset was seven, so we used a seven-fold cross-validation strategy. Each session consisted of six plans for training and one for testing. Table 2 shows the results of the experiment on the UOS dataset.

**Table 2.** Class-wise accuracy comparison on different GNN models on the UOS data set.

| GNN Model | AGG | Objects | Wall | Window | Door | Stair | Lift | Room | Hallway | X-Room | Overall |
|---|---|---|---|---|---|---|---|---|---|---|---|
| GCN | · | 0.7286 | 0.6829 | 0.6286 | 0.6314 | 0.7643 | 0.8043 | 0.56 | 0.4143 | 0.4357 | 0.6843 |
| GIN | MEAN | 0.7014 | 0.7614 | 0.5629 | 0.67 | 0.7957 | 0.7071 | 0.4886 | 0.3643 | 0.4729 | 0.71 |
| GIN | MAX | 0.7529 | 0.7857 | 0.7114 | 0.73 | 0.7186 | 0.4686 | 0.5671 | 0.49 | 0.4743 | 0.7457 |
| GIN | SUM | 0.8014 | 0.87 | 0.84 | 0.7914 | 0.7786 | 0.7414 | 0.6757 | 0.65 | 0.5086 | 0.8329 |
| GraphSAGE | POOL | 0.8371 | 0.87 | 0.8357 | 0.7971 | 0.8514 | 0.6714 | 0.7571 | 0.6214 | 0.5614 | 0.8429 |
| DWGNN | MEAN | 0.8626 | 0.8879 | 0.8526 | 0.8256 | 0.8857 | 0.6525 | 0.8382 | 0.7712 | 0.6686 | 0.8658 |
| DWGNN | SUM | 0.8633 | 0.8916 | 0.8684 | 0.8274 | 0.9103 | 0.8454 | 0.8067 | 0.8087 | 0.7142 | 0.8764 |
| DWGNN | MAX | 0.8644 | 0.8943 | 0.861 | 0.8165 | 0.9191 | 0.7323 | 0.8293 | 0.8155 | 0.7353 | 0.8765 |
| DWGNN | LSTM | 0.8665 | 0.9178 | 0.923 | 0.8661 | **0.9457** | 0.7875 | 0.8406 | **0.8385** | 0.7765 | 0.9072 |
| GraphSAGE | LSTM | **0.908** | **0.9308** | **0.9152** | **0.8847** | 0.9318 | **0.9206** | **0.8599** | 0.7951 | **0.8255** | **0.9184** |

The overall accuracy score was lower than that of the CubiCasa dataset. The spatial element class underperformed compared to the CubiCasa dataset since non-spatial classes in the UOS dataset had large doors and lifts, whose area was large and could be added to the spatial class. Like the CubiCasa dataset, transductive learning-based models underperformed compared to the inductive learning-based models. Unlike the previous experiment, GraphSAGE with the LSTM aggregator was not ranked first place in every element class, and for stairs and hallways, DWGNN did better than GraphSAGE with LSTM (see Table 2). This is because the shapes of stair elements are more defined compared to that of CubiCasa, and DWGNN could generalize the structured set of polygons and find the patterns of their formation better than GraphSAGE. For the hallways, they tend to be linked to many other elements with respective distances, and thus make DWGNN easy to generalize regarding the characteristics of hallways by taking the attention values into account (shown in Figure 6).

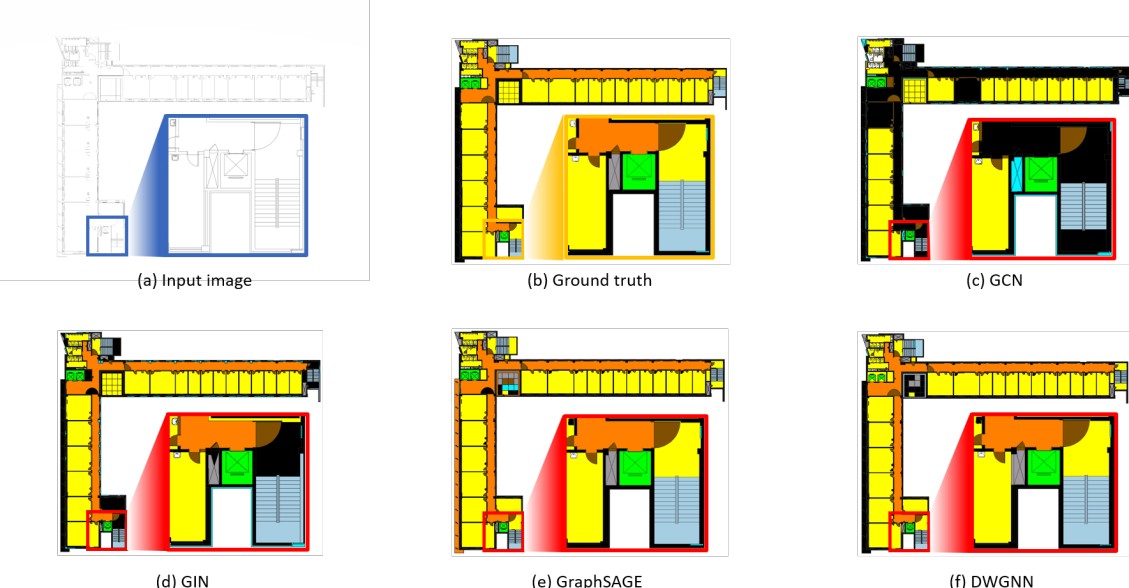

(a) Input image  (b) Ground truth  (c) GCN

(d) GIN  (e) GraphSAGE  (f) DWGNN

**Figure 6.** Visualized results of classification on UOS dataset.

As the number of plans in the UOS dataset was limited, the generalization of the characteristics of classes was difficult. If a GNN model has a node that has never been seen before, the node will not only affect itself, but also the neighboring nodes up to K hops

away. This occurs because GNN aggregates the feature of a much wider range of nodes as the number of layers increases. In addition, the GNN model may simply memorize the training dataset as the number of plans is limited. To alleviate these problems, we augmented the UOS dataset using an affine transformation. For all points in the set of floor plan polygons, a point was scaled about the origin with a scale factor of 0.7, then flipped over the y-axis. After that, we rotated the polygons 90 degrees counterclockwise (see Figure 7). The transformation formula is as follows.

$$
\begin{bmatrix} x' \\ y' \\ 1 \end{bmatrix} = \begin{bmatrix} \cos 90° & -\sin 90° & 0 \\ \sin 90° & \cos 90° & 0 \\ 0 & 0 & 1 \end{bmatrix} \begin{bmatrix} 0.7 & 0 & 0 \\ 0 & -0.7 & 0 \\ 0 & 0 & 1 \end{bmatrix} \begin{bmatrix} x \\ y \\ 1 \end{bmatrix} \tag{9}
$$

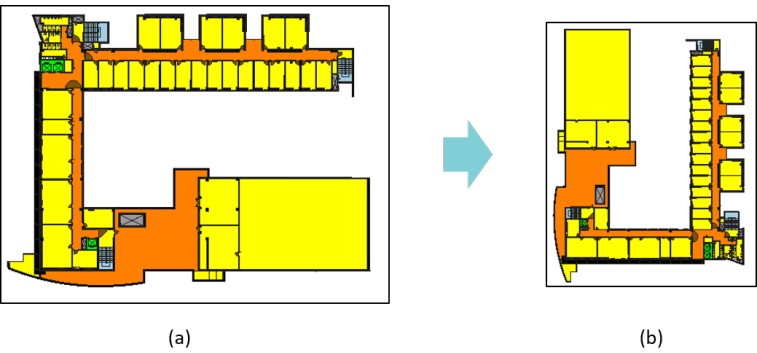

(a)        (b)

**Figure 7.** Example of data augmentation. The original plan (**a**) gets transformed by Equation (9), and returns an augmented plan (**b**).

We formed a new dataset UOS-aug consisting of the seven augmented plans with the original plans from UOS. Because classification performance was improved through data augmentation, we can derive that the results of the GNN model are invariant to scale and rotation. In addition, this proves that the GNN model learns a pattern of updating the embedding vectors of that node in relationship with the neighbors of each node, rather than memorizing the structure of the drawing. The results are shown in Table 3.

**Table 3.** Class-wise accuracy comparison on different GNN models on the UOS-aug data set.

| GNN Model | AGG | Objects | Wall | Window | Door | Stair | Lift | Room | Hallway | X-Room | Overall |
|---|---|---|---|---|---|---|---|---|---|---|---|
| GCN | · | 0.8812 | 0.7898 | 0.7295 | 0.7565 | 0.8516 | 0.9243 | 0.6508 | 0.6619 | 0.6255 | 0.7822 |
| GIN | MEAN | 0.886 | 0.8865 | 0.8656 | 0.8208 | 0.95 | 0.8406 | 0.7603 | 0.5446 | 0.8215 | 0.8752 |
| GIN | MAX | 0.9014 | 0.9373 | 0.9094 | 0.906 | 0.9706 | 0.9348 | 0.9026 | 0.8734 | 0.8822 | 0.925 |
| DWGNN | MEAN | 0.9334 | 0.9516 | 0.9472 | 0.9092 | 0.9705 | 0.9405 | 0.9289 | 0.8598 | 0.8862 | 0.9446 |
| DWGNN | SUM | 0.9602 | 0.9608 | 0.9511 | 0.9301 | 0.9728 | 0.9311 | 0.9469 | 0.9088 | 0.8966 | 0.955 |
| DWGNN | MAX | 0.9612 | 0.9683 | 0.9588 | 0.9398 | 0.9803 | 0.9659 | 0.9531 | 0.9208 | 0.9189 | 0.9628 |
| GIN | SUM | 0.964 | 0.9731 | 0.9679 | 0.9286 | 0.9766 | 0.9531 | 0.9661 | 0.9358 | 0.9004 | 0.9658 |
| GraphSAGE | POOL | 0.9696 | 0.9727 | 0.9689 | 0.9485 | 0.974 | 0.9617 | 0.9459 | 0.8946 | 0.933 | 0.9681 |
| DWGNN | LSTM | 0.9627 | 0.978 | 0.9733 | 0.9474 | **0.9848** | 0.9507 | 0.9488 | **0.9401** | 0.9478 | 0.9716 |
| GraphSAGE | LSTM | **0.9762** | **0.9827** | **0.9802** | **0.9646** | 0.9796 | **0.9869** | **0.9553** | 0.9124 | **0.9747** | **0.9788** |

The results improved compared to Table 2. Though the augmented plans have gone through many changes, they worked in a complementary manner with the original plans, which means that the GNN models are invariant to scale and rotation. This proves that the GNN models classify their nodes using the relationship and patterns among nodes and features within each graph, not the formation and arrangement of nodes.

## 5. Discussion

The contributions of our work are as follows. First, we developed a raster to vectorization process for floor plan images independent of the drawing style. With appropriate image pre-processing methods, it can convert any type of floor plan image into polygon vector data. By vectorizing the floor plan image before pixel segmentation, we were able to capture not only structural elements, but symbols and spatial elements without losing shape information. Second, to classify the polygons, we employed the Graph Neural Network approach. The GNN models are invariant to scale and rotation since GNN takes input as a graph, and the graph data structure has no fixed permutation of nodes. Utilizing GNN makes the framework robust and easy to generalize floor plan datasets of any style. Third, we defined the need for inductive learning GNN models for floor plan element classification tasks and, among many GNN models, we chose an appropriate one (GraphSAGE). Further, we developed a new GNN model taking the distance weight value into account in the message passing process using the softmin function.

While the results showed that our framework can detect and classify multi-labeled floor plan elements, a few limitations were derived as follows. The features currently used in the feature matrix of polygons are significant, but if we use additional feature information that fully describes individual polygons among different types of floor plan elements, it would be possible to do additional classification. The proposed framework outputs the result in a vector format, which facilitates its use in additional research or real-world applications. For example, Zeng et al. [10] demonstrated the 3D models of the results from their method, and the output of the proposed framework is already vector-type data, making it even easier for 3D modeling.

Unlike CNN-based models, which are robust to noisy images, the application of the proposed framework to noisy or low-resolution images is difficult. Especially in the image pre-processing phase, the output is highly dependent on the noise and the resolution. For example, if the pixel values of the symbol are uneven due to low resolution, doors tend to lose the exact arc line and fail to get converted into a polygon. To overcome these limitations, an image generation model can be applied and used in the pre-processing step. However, due to the nature of the generative model, it is difficult to expect detailed improvement at the pixel level. In addition, our framework does not utilize text information in the image, thus rendering impossible the use of semantic information, that explicitly indicates the nature of each object.

In most experiments, the DWGNN showed slightly lower accuracy than GraphSAGE. It is because, on the RAG conversion stage, the node of the graph corresponds to the centroids of the polygons and the weight value is calculated between the coordinates of the pair of nodes, thus preventing them from holding the shape information of the polygons. Especially for walls or outer space, most of the node coordinates that represent polygons are often situated where the actual polygon is not located. To alleviate this, DWGNN uses the softmin function to assign the attention values; however, the meaningless edge features still prevent the model from being trained and predicting the classes correctly. With the nature of DWGNN, we think that it can be an appropriate model for solving combinatorial optimization problems in spatial networks, such as the traveling salesman problem or vehicular routing problems, rather than for graphs with polygon nodes.

## 6. Conclusions

This paper presents a new framework for extracting and classifying the elements in a floor plan. Unlike previous approaches that first segment the floor plan image, our method vectorizes the floor plan images and converts the polygon set into an RAG. The model then employs a GNN to classify the nodes in the graph according to their unique features and neighborhood relationship. Inductive learning was conducted on the floor plan graphs in order to predict completely unseen graphs. Our framework classifies not only basic element and symbol classes but also spatial elements such as rooms, with resulting vector format outputs to minimize the abstraction and loss of shape information. To evaluate the

performance of the proposed framework, we performed experiments on two floor plan datasets with different areas and distributions and one data augmented dataset. Results showed high accuracy rate on the classification task with the expressive power of the final output. By comparing various GNN models, we also found that inductive learning-based GNN models outperform transductive learning-based models. As further research, we will find a way to handle low-resolution floor plan images and improve the classification performance by extracting additional features.

**Author Contributions:** Conceptualization, Jaeyoung Song; methodology, Jaeyoung Song; software, Jaeyoung Song; validation, Jaeyoung Song; investigation, Jaeyoung Song; data curation, Jaeyoung Song; writing—original draft preparation, Jaeyoung Song; writing—review and editing, Jaeyoung Song and Kiyun Yu; visualization, Jaeyoung Song; supervision, Kiyun Yu; project administration, Kiyun Yu; funding acquisition, Kiyun Yu. All authors have read and agreed to the published version of the manuscript.

**Funding:** This research was supported by a grant(21NSIP-B135746-05) from National Spatial Information Research Program (NSIP) funded by Ministry of Land, Infrastructure and Transport of Korean government.

**Data Availability Statement:** No applicable.

**Conflicts of Interest:** The authors declare no conflict of interest.

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
