# Peer review of "Framework for Indoor Elements Classification via Inductive Learning on Floor Plan Graphs"

_ijgi, doi:10.3390/ijgi10020097_

Round 1

Reviewer 1 Report

This paper presents a framework to classify floor plan elements by transforming them to vector format and using GNN.  The method is sound, relatively novel (at least in this area), and good classification results are achieved. Additional merit of this paper is a new UOS dataset, which can benefit the community and may be of interest to the general image classification community. The paper is well-written and easy to read.

GNN are described in detail, which is too much for CS, but is suitable for the Civil Engineering research. It is nice to see advanced ANNs used in more fields.

Detailed comments:

l. 10-11 Please define what a 'floor plan' is for readers who are not from this field.

l. 19-20 Why is this feature valuable? Where does it help?

l. 109 Why capitalization in 'Graph'?

p.5 last paragraph: Centroid is usually a term used for cluster of points in metric space; stating that this is a polygon centroid here might help.

l.182 Is this selection empirical?

l.262 Do you mean edge weights?

p.9 Imho, what is needed here is a toy example of simple layout (say, one room), and its resulting graph with edge weights.

l.305-307 Please supply the URL for downloading the dataset (or for sending requests to use the dataset for academic purposes). Kaggle is also an option.

Author Response

Thank you for your comments. 

We sincerely tried to revise our manuscript according to your advice.

Please see the attached file below to see the responses to your comments.

Reviewer 2 Report

Ms. Ref. No.: ijgi-1107544
Title: Framework for Indoor Elements Classification via Inductive Learning on Floor Plan Graphs

Overview and general recommendation:

The manuscript presents a new approach to classify and represent in vector form elements in building floor plans. The text is clear, the approach detailed and scientifically sound. Figures and tables are also clear to the reader. The conclusions are based on the results. The only major concern is the mismatch between the purpose of the manuscript (computer vision; machine learning; pattern recognition) and the journal audience (geo-information). In this case, the relation to geo-information field is vague if not absent. A minor recommendation is to avoid lump sum citations and critically review each one individually. Therefore, the recommendation is to transfer this manuscript to a journal where the audience would appreciate its contribution to computer science.

Author Response

Thank you for your concern.

The purpose of this paper is to detect and classify indoor objects in a floor plan automatically by using Graph Neural Network approaches. Even though it looks a bit different from the purpose of the journal, using graphs for indoor analysis is very popular in the field of GIS and geo-informatics. Indoor graphs help us model an indoor layout as each entity (node) can share the information using links among them. The paper utilizes a region adjacency graph to classify the class of each element, by sharing and aggregating the information of indoor elements. In my humble opinion, we can further use the method and result of this paper on GIS sub-field such as indoor navigation systems or object placement/installation.

We deleted several lump sum citations in section 2 as you recommended.

We will further research studies related to the indoor GIS field by utilizing and developing the method we proposed.

Reviewer 3 Report

Although in most of the cases the results of the proposed DWGNN are not the best, this is an interesting and well written paper. Several things are, however, not fully clear and I would suggest improving/explaining them:

  • In Figure 1 RAG is presented. I do not understand why there is a node in the middle on the right. Which polygon does it represent? Is it a centroid of some polygon?
  • Content of section 3.1 (in particular second paragraph describing vectorization process) as well as Figure 2 are not clear.
  • In lines 219 and 220 there is m symbol while in equation (2) there is h instead.
  • What is object class in UOS dataset?
  • In lines 328-331 hyper-parameters are described. How they were selected? Was some validatiion/cross-validation procedure used?

Author Response

(The authors gave the same response as above.)
